# Understanding sociocultural influences in sexual health promotion and HIV protection among Latinx sexually minoritized men: A qualitative study

Lisvel A. Matos[ID][1]*, Sarah E. Janek[ID][1], Jasmine Salas[1], Caroline Munoz[1], Michael V. Relf[1,2], Rosa Gonzalez-Guarda[1]

1 School of Nursing, Duke University, Durham, North Carolina, United States of America, 2 Global Health Institute, Duke University, Durham, North Carolina, United States of America

* lisvel.matos@duke.edu

## Abstract

In the United States, Latinx individuals account for 18% of the overall population, yet sexual minoritized men (SMM) within this demographic disproportionately represent 30% of new HIV diagnoses among all SMM. Despite the availability of highly effective HIV prevention strategies such as Pre- exposure Prophylaxis (PrEP), there are marked inequities in access and utilization of these strategies among Latinx SMM. Sociocultural factors and the experiences of Latinx SMM shape beliefs about sexual health promotion and influence HIV self-protective behavior (e.g., the correct and consistent use of condoms, regular engagement in HIV testing, and PrEP use) in this group. A descriptive, qualitative study, using thematic analysis, was designed to describe the sociocultural barriers and facilitators that influence sexual health promotion and HIV self-protection among Latinx SMM. A convenience sample of 15 Latinx SMM was recruited from an ongoing longitudinal study and online for individual interview, which were conducted virtually between October 2020 and October 2021. Five themes emerged from the data: 1) prevention is better than to cure; 2) cultural and religious norms create a culture of silence around sexual health; 3) lack of information and misinformation leading to self-reliance for sexual health protection; 4) growing wiser: maturity's impact on sexual health and relationships; and 5) unjust and dehumanizing sexual health care. The findings from this study highlighted that stigma and structural factors shape the opportunities of Latinx SMM to engage in sexual health promotion and HIV self-protection. These findings emphasize the need for focus on addressing structural barriers such as sexual health education, access to care, and healthcare provider bias to improve sexual health outcomes among Latinx SMM.

## Introduction

Latinx individuals represent approximately 19% of the United States (U.S.) population, yet they accounted for nearly one-third of new HIV infections in 2022 [1,2]. The term Latinx refers to individuals of Latin America origin or descent, regardless of whether Spanish is their

**Data availability statement:** Coded interview data can be found in the Qualitative Data Repository (QDR) at https://doi.org/10.5064/F6YQVSMZ. Full anonymized interview transcripts have not been shared in the repository due to a lack of informed consent for data sharing and the highly specific information shared by participants, which increases the risk of deductive disclosure. Requests for anonymized transcripts can be sent to ResearchServiceDesk@duke.edu and will be reviewed by appropriate offices at Duke University.

**Funding:** This research was supported by the Holditch-Davis PhD Student Research Study Award and the Duke-Margolis Scholars Program. The funders had no role in the design of the study, data collection and analysis, decision to publish, or preparation of the manuscript.

**Competing interests:** The authors have declared that no competing interests exist.

primary language [3,4]. This term embraces the linguistic diversity of the Latin American diaspora, which includes Spanish, indigenous languages, French, and Portuguese [3,5]. Latinx sexual minoritized men (SMM) in the U.S. are most profoundly affected by HIV, comprising 77% of new infections among Latinx individuals and 36% of new diagnoses among SMM in 2022 [2]. Although overall new HIV diagnoses declined by 15% among SMM from 2010 to 2022, the rates of new HIV diagnoses among Latinx SMM increased by 24% in that same time frame, indicating inequities in prevention efforts [6]. Approved by the FDA in 2012, HIV pre-exposure prophylaxis (PrEP) is 99% effective at preventing HIV transmission but remains underutilized among Latinx SMM. For example, data from a recent survey showed that only 6% of Latinx individuals who could benefit from PrEP, most of whom are SMM, are utilizing PrEP compared to 25% of non-Latinx Whites [7,8]. These inequities in PrEP access and use highlight a critical gap in HIV prevention among this group.

The introduction of PrEP has transformed the HIV prevention landscape, shifting from relying solely on traditional behavioral methods like condom use to a biomedical approach that offers individuals greater control over protecting themselves against HIV [9]. This critical change has not only redefined self-protection strategies against HIV but also prompted researchers to broaden their focus when understanding factors that influence the uptake of HIV prevention behaviors [9]. A recent study demonstrated that PrEP use and adherence of PrEP is associated with increased engagement in sexual health services and heightened awareness of overall well-being among users, indicating a broader impact on sexual health beyond HIV prevention alone [9]. Findings from a qualitative study found that PrEP use increased self-awareness of sexual risk-taking behaviors and sexual health among Latinx SMM [10].

The World Health Organization (WHO) describes sexual health as "a state of physical, emotional, mental and social well-being in relation to sexuality; it is not merely the absence of disease, dysfunction or infirmity" [11]. This definition also includes the promotion of safe, consensual, and satisfying sexual experiences, and recognizes the influence of socio-economic, cultural, and environmental factors on sexual health and protective behaviors [11]. This broader understanding of sexual health is highly relevant for Latinx SMM, who navigate a complex web of influences on their sexual well-being.

Essential aspects of healthy human sexuality such as communication about sex, positive sexual identity, sexual health care, and sexual autonomy are often constrained by social norms that converge in unique ways within Latinx communities [12,13]. The pervasive influences of machismo (strong sense of masculine pride), sexual silence (the complete suppression of openly discussing sex and sexuality), and religion [14–16] create heteronormative, sexually conservative environments. These environments stigmatize and constrain Latinx SMM's ability to openly express their sexual identities [17]. They also impede healthy sexual exploration, access to accurate information, and access to sexual healthcare because they perpetuate stigma and discrimination and create a culture of silence around sexuality [15,18,19]. As a result, Latinx SMM may experience misinformation, fear, and shame, further discouraging them from seeking necessary healthcare services and openly discussing their sexual health [20–22]. In addition, structural challenges, including healthcare provider bias, language barriers, and lack of health insurance limit access to essential sexual health care services like PrEP among Latinx SMM [23–25]. Despite these challenges, there are Latinx SMM who successfully navigate this complex landscape to engage in sexual health promotion. However, less is known about the strategies and experiences that enable Latinx SMM to promote their sexual health, particularly since the introduction of PrEP as an HIV prevention tool. Gaining insight into the barriers they face and the facilitators they leverage is crucial for developing targeted interventions and improving access to sexual health resources in this population.

This study specifically addresses the gap in understanding how Latinx SMM navigate the complexities of sexual health promotion and HIV self-protection within their unique socio-cultural environments, an area that has been underexplored in current research. Given the wide range of options available for HIV prevention, this study aims to: 1) describe sexual health promotion and HIV self-protection among Latinx SMM, including how they perceive sexual health and the specific behaviors they adopt for HIV self-protection; and 2) describe the sociocultural barriers and facilitators that influence sexual health promotion and HIV self-protection among Latinx SMM. Sexual health promotion refers to the strategies and activities aimed at enhancing sexual health and well-being by preventing sexually transmit-ted infections (STIs) including HIV, improving sexual health knowledge and behaviors, and increasing access to sexual health services [11]. We define HIV self-protection as the actions and practices individuals undertake to reduce their risk of acquiring HIV, including the correct and consistent use of condoms, regular engagement in HIV testing, practicing safer sex, and utilizing PrEP. By investigating the experiences of Latinx SMM with sexual health promotion and HIV-self-protection, this study aims to provide a greater understanding of sexual healthcare needs among Latinx SMM. In this study, the term 'Latinx' refers specifically to the Spanish- or English-speaking sub-population of individuals who self-identified as being born in Latin America or as U.S.-born with Latin American ancestry.

## Methods

### Study design

This study used a qualitative descriptive design [26] to explore sexual health promotion and HIV self-protection among cisgender Latinx SMM living across the U.S. Individual in-depth qualitative interviews were conducted to inquire about participants' perceptions of sexual health promotion. Additionally, we inquired about socio-cultural barriers and facilitators of specific HIV and STI prevention strategies such as condom use, PrEP use, and partner negotiation.

### Participants

Between October 2020 and July 2021, a purposive sample of 15 Latinx SMM was recruited. Our study specifically focuses on a sub-population of Latinx individuals, as all participants reported Latin Amsssssserican ancestry and were either Spanish- or English-speaking. The eligibility criteria included: self-identifying as being Latinx (U.S. born or foreign-born), cisgender male, English or Spanish-speaking, aged 18 and older, and living without HIV. Par-ticipants were excluded if they lived outside the United States, were living with HIV, or were not fluent in Spanish or English.

Recruitment for this study was conducted through three methods. First, participants who met our inclusion criteria were purposively sampled from an ongoing study of Latinx young adults (1R01MD012249; PI: Gonzalez-Guarda). The SER Hispano study is an ongoing study on the health of young adult immigrants (18–44 years) in the southeastern region of the US (N=391). Participants who met eligibility criteria and agreed to be contacted for future studies were provided information about this study. Eleven eligible participants were identified from The SER Hispano study and consented to be contacted regarding future research, but only three participants enrolled in this study. Second, recruitment messages were posted on Latinx Facebook groups, Reddit threads for Latinx or SMM, and Craigslist. Participants who were recruited online contacted researchers by email or phone and were screened for study eligi-bility. Snowball sampling techniques were also used [27]. For example, Latinx SMM who had already enrolled in the study were asked to inform friends who met study inclusion criteria.

All participants were screened by one of four bicultural, bilingual researchers that were a part of this study. A total of 20 participants were screened for this study, and 15 individuals ultimately enrolled. Of the five individuals who did not enroll, two were ineligible due to living with HIV, two were ineligible because they resided outside of the U.S., and one was lost to follow-up during the recruitment process. Of the 15 enrolled participants, three were recruited from an ongoing study, 11 recruited through social media, and one recruited using snowball sampling techniques. Participants received a $30 Visa gift card for their participation.

## Setting

Interviews were conducted virtually via WebEx, although four participants experienced connectivity issues and were interviewed via phone call. Interviewers and participants were in a private location during the interview.

## Data collection

All interviewers participated in a training on effective interviewing strategies by the senior author. A semi-structured interview guide was used to explore the meaning of sexual health and the various strategies utilized to prevent HIV among Latinx SMM. Additionally, the interview guide explored how interpersonal-, provider-, and sociocultural-level factors influenced sexual health promotion and HIV prevention. Sample questions included: "What does Sexual health promotion mean to you"; "What types of methods or activities do you engage in to protect yourself from HIV?"; and "In what ways do you feel like your partner(s) are helpful/not helpful when making decisions about your sexual health?". The guide was reviewed by experts in the field of HIV prevention, and gay Latinx men community members.

Interviews were conducted by bilingual, bicultural researchers (L.M., J.S., C.M., R.G.G.). Participants were encouraged to respond in whichever language felt most comfortable to them throughout the interview. Most interviews (n=8) were conducted in English. Participants were also asked to complete a brief demographic survey after the interview was completed. Each individual interview lasted between 45 and 65 minutes, with an average duration of 52 minutes, and was audio-recorded for transcription and analysis.

## Study materials

All materials used in this study were initially translated from English to Spanish by a bilingual team member. To ensure accuracy and cultural and linguistic relevance, the translated materials were then backtranslated from Spanish to English by another bilingual team member who was not involved in the initial translation process. Any discrepancies between the original and back-translated versions were resolved through discussion and consensus among the research team [28].

## Human subject protection and data management

The Institutional Review Board of Duke University approved all study materials (IRB# Pro00106283). Prior to participation in interviews, all participants verbally provided informed consent, acknowledging their understanding of the study's purposes and the voluntary nature of their involvement. Interviews were digitally recorded and stored on a secured folder. The audio recordings were then transcribed verbatim by a professional transcription service that complies with confidentiality agreements to further protect participant privacy. To maintain the integrity and nuanced meaning of the data, all transcripts were kept in their original language. This ensures that the cultural and linguistic contexts of participants' responses are preserved, allowing for a more accurate analysis. Once the transcripts were prepared, they were

verified for accuracy by the lead author. Access to these transcripts is restricted to authorized research team members only, who are required to adhere to strict data confidentiality protocols as outlined by the IRB-approved study guidelines.

## Data analysis

NVivo 13 [29] was used for the management and analysis of interview data. Thematic analysis techniques were applied to analyze transcripts in their original language with bilingual coders fluent in both English and Spanish [30]. An initial analysis of the first five interviews was conducted by the primary researcher and two team members (J.S., C.M.). Researchers immersed themselves in the data by reading and re-reading the transcripts without coding [31]. Subsequently, they highlighted meaningful units, relevant information, and key ideas present in the data [30]. Open coding was then employed to label highlighted text that captured their interpretation of the underlying meaning of participant comments. A preliminary codebook was developed based on the first five transcripts. The codebook contained codes, their content descriptions, and a brief example for reference [32].

The remaining transcripts were independently coded by the primary researcher as interviews were completed, using the same approach as described above. Throughout the coding process, additional codes that emerged from the data were added as needed to capture the richness and complexity of the data. These newly identified codes were also applied retrospectively to the first five transcripts to ensure consistency and a comprehensive analysis across all cases. After the data were coded, the primary researcher analyzed codes to generate a list of categories, which were then grouped into themes [33]. Themes were identified based on patterns, similarities, and differences across the data. Throughout the data analysis process, the primary researcher engaged in weekly discussions with senior researcher and qualitative expert on the team (R.G.G.) to review coding refine identified themes that emerged from the data. To further assess the reliability of the coding process, 20% of the transcripts were randomly selected and independently coded by (S.J.) and (R.G.G.). Any discrepancies in coding were resolved through discussion and consensus among the research team. The major themes that are reported in this paper were areas responding to our research question where saturation was reached.

Trustworthiness of the findings were ensured using several methods. To establish credibility, peer debriefing was used at several stages of this study. Researchers conducted regular peer debriefings during the recruitment interview and analysis stages [32,34]. Member checking was conducted by having participants review and respond to the findings, confirming their accuracy and providing recommendations. To establish transferability, researchers used field notes to provide detailed accounts of their experiences during data collection [34]. Interviews were analyzed in their original language to capture the full meaning of the text, ensuring that the transcripts accurately reflected the participants' experiences and enhanced the transferability of the findings [35,36]. To establish dependability, a researcher not involved in the study assessed the research process and outcomes, ensuring the findings, interpretations, and conclusions were accurately supported by the data [34]. To establish confirmability an audit trail was maintained to document all research processes and decisions. [34].

## Reflexivity

As a research team, it was crucial for us to minimize the influence of our own biases and assumptions throughout the research process. Most of the research team identified as Latinx, cisgender females, and heterosexual. The team also included LGBTQ+ members as well as individuals who were non-Latinx White. All the interviews were conducted by the Latinx cisgender heterosexual researchers on our team. Prior to conducting the interviews, the

interviewer(s) acknowledged that even though they have a shared cultural understanding with participants, they may not fully comprehend the experiences of Latinx SMM. The interviewer(s) highlighted their commitment to listening and learning from the unique perspectives and experiences of participants. This conversation was aimed at fostering mutual understanding and establishing a foundation of trust and respect between the researchers and participants. Participants were also provided with the opportunity to ask questions and share their own reflections on how the interviewer(s) positionality might impact their participation and the overall research process at the end of the interview.

## Results

The sociodemographic characteristics for participants (n=15) are presented in Table 1. Participants resided in North Carolina (n=7), Texas (n= 3), California (n=3), Florida (n=1), and Minnesota (n= 1) at the time of the study (N=15). The participants' mean age was 31.4 years

**Table 1. Participant Sociodemographic Characteristics (N=15).**

| Characteristics | N (%) |
| --- | --- |
| **Age (Mean)** | 31.4 |
| **Nativity, Foreign-born** | 8 (53.3) |
| **Ancestry** | |
| Central American (Mexico, Nicaragua, Guatemala, Honduras) | 8 (53.3) |
| Caribbean (Dominican Republic, Puerto Rico) | 4 (26.6) |
| South America (Chile, Columbia, Venezuela) | 3 (20.0) |
| **Employment** | |
| Full-Time | 9 (60) |
| Part-time | 1 |
| Self-employed | 3 (20) |
| Unemployed | 1 |
| Student | 1 |
| **Annual Income** | |
| <$10,000 | 1 |
| $10,000-$29,999 | 1 |
| $30,000-$49,999 | 4 |
| $50,000-$75,000 | 7 |
| >$75,000 | 2 |
| **Education** | |
| Some college, no degree | 6 |
| Associate degree (e.g., AA, AS) | 2 |
| Bachelor's degree (e.g., BA, BS) | 6 |
| Master's degree (e.g., MA, MS, MEd) | 1 |
| **Relationship Status** | |
| Single | 7 |
| In a relationship (closed) | 5 |
| In a Relationship (open) | 3 |
| **PrEP use** | |
| Current | 6 |
| Former | 2 |
| No PrEP use Hx | 7 |

(range 24–43). More than half of the participants (n=8) were born outside of the U.S. Most participants were employed (n=13) and reported having an associate degree or higher (n=9). Nearly all participants (n=14) were aware of PrEP and nearly half (n=7) were using PrEP. Two participants had a history of PrEP use and were not taking PrEP at the time of the interview. Individual participant profiles are provided in Table 2, detailing participant number, age, relationship status, sexual identity, preferred language, country of birth, and PrEP use.

We identified five major themes when exploring the sociocultural barriers and facilitators of sexual health promotion HIV self-protective behaviors. The identified themes were: 1) prevention is better than to cure; 2) cultural and religious norms create a culture of silence around sexual health; 3) lack of information and misinformation leading to self-reliance for sexual health protection; 4) growing wiser: maturity's impact on sexual health and relationships; and 5) unjust and dehumanizing sexual health care. We detail related themes and subthemes below, presenting exemplar quotes in the original language from the transcripts to preserve the linguistic and cultural authenticity of the participants' experiences [35,36]. For quotes originally in Spanish, English translations are also provided in italics.

## Prevención es mejor que curar/prevention is better than to cure

Throughout the interviews, participants primarily described sexual health promotion as having sex responsibly (e.g., using condoms) and the prevention of STIs including HIV. Participants also described sexual health as a holistic concept beyond disease prevention and included physical, emotional, and psychological well-being. This perspective informed their approach to HIV self-protection, which was seen not as a singular task but as part of a broader commitment to health and described by both Spanish and English-speaking participants.

"Bueno, pues… estoy a punto de cumplir treinta años, y me estoy haciendo plenamente responsable de mi salud individual entonces e encontrado que. que prevenir es mejor que curar. Entonces implica en todo… en alimentación, en ejercicio, en… no solamente en el área de cuidado de salud sexual, si no en general. Sabes comiendo menos sodio, menos azucares, menos comidas procesadas. Entonces encuentro como mejor vía la prevención que el

**Table 2. Participant Profiles (N=15).**

| Participant Number | Age | Relationship Status | Sexual Identity | Preferred Language | Country of Birth (ancestry) | PrEP User |
|---|---|---|---|---|---|---|
| #01 | 29 | Single | Gay | Spanish | Honduras | No |
| #02 | 40 | Open Relationship | Gay | English | Nicaragua | Yes |
| #03 | 24 | Single | Gay | English | U.S. (Mexico) | No |
| #04 | 29 | Monogamous Relationship | Gay | English | Mexico | No (former user) |
| #05 | 34 | Monogamous Relationship | Gay | Spanish | Mexico | No |
| #06 | 27 | Single | Gay | English | U.S. (Mexico) | Yes |
| #07 | 32 | Polyamorous Relationship | Bisexual | English | U.S. (Puerto Rico) | No |
| #08 | 26 | Open Relationship | Gay | Spanish | Chile | Yes |
| #09 | 32 | Single | Gay | English | U.S. (Mexico) | No |
| #10 | 26 | Monogamous Relationship | Gay | Spanish | Colombia | Yes |
| #11 | 26 | Monogamous Relationship | Gay | English | U.S. (Venezuela) | No |
| #12 | 34 | Single | Gay | English | Guatemala | No (former user) |
| #13 | 29 | Single | Gay | English | U.S. (Dominican Republic) | No |
| #14 | 43 | Open Relationship | Gay | English | U.S. Territory (Puerto Rico) | Yes |
| #15 | 40 | Single | Gay | Spanish | Dominican Republic | Yes |

tratamiento de cualquier enfermedad. [*Well, I'm about to turn thirty, and I'm becoming fully responsible for my individual health, so I've found that prevention is better than to cure. So, it involves everything... in terms of diet, exercise, not only in the area of sexual health care, but in general. You know, eating less sodium, less sugar, less processed foods. So, I find prevention to be a better approach than treating any disease.*]" (Participant #04, 29 year-old, single)

"Being sexually healthy means taking care of yourself in every aspect, not just when it comes to HIV but all STIs, and even beyond that, making sure you're mentally and emotionally okay too" (Participant, #13, 29 year-old, single)

**HIV testing.** HIV self-protection was a sexual health priority among participants. As one participant expressed, "*I think, for me like the scariest part about umm being gay and um having sex with other men is the risk of HIV.*" (Participant #03, age 24, single). Participants incorporated HIV testing as a routine part of health promotion, undergoing tests annually regardless of relationship status. PrEP users reported testing every three months as part of their regular follow-up care. Participants reported different frequencies of testing, with some testing more frequently depending on their sexual activity or if they were symptomatic for any STIs. While discussions did not specifically include HIV self-testing, participants often referred to going to a clinic or seeing a healthcare provider for testing. One participant (age 29, single) had never been tested for HIV nor spoken to a healthcare provider about testing. He expressed considering getting tested in the past but decided against it because he had no HIV symptoms.

**Condom use.** Condoms were also described as an important tool for HIV self-protection. However, participants perceived a general decline in condom use within the gay community. This led them to anticipate that sexual partners would prefer condomless sex, which was highlighted by one participant who stated, "*everyone likes to do more of that skin-to-skin contact*" (participant #13, age 29, single). Despite the perceived preferences of having condomless sex, several participants were prepared to engage in negotiations over condom use with partners, with another participant comparing this process to "*mental acrobatics*" (participant #03, age 24, single). He also noted that condom negotiation was particularly challenging for oral sex, where using condom use is not commonly accepted as part of "*sex culture*".

While condoms were recognized as a crucial tool for HIV self-protection and participants were prepared to negotiate their use, this did not come without challenges. Deviating from the previous participant's perspective, one participant highlighted the impact of condoms on sexual pleasure, "*I've never...never...never put a condom to suck someone off...never. It takes the joy of that, that experience*" (participant #12, age 34, single). Other participants echoed this sentiment expressing that condoms are uncomfortable and take away from the sexual experience, "*the use of condoms is a little bit uncomfortable from the you know… just the process of it. Like it's just the elastic, the smell, it just kind of takes away from that*" (participant #14, age 40, open relationship). Furthermore, several participants shared that condoms interfere with sexual function, making it more difficult for them to maintain an erection. These factors challenged the consistent use of condoms for HIV self-protection among participants.

**PrEP.** PrEP awareness was high among participants; however, knowledge about PrEP varied. As expected, previous PrEP users demonstrated a greater understanding of PrEP than those who had not used PrEP previously. One participant, had heard of PrEP but did not know what the medication was used for until this interview,

"Porque yo lo escuchado porque yo tengo una aplicación que se llama Grindr… ponen siempre en su perfil que están en PrEP…. Pero yo nunca me pregunte que que era PrEP, y

como era [*Because I had heard about it, because I have an app called Grindr... they always put on their profile that they are on PrEP... But I never wondered what PrEP was, and how it worked*]" (Participant #01, age 29, single).

Participants who were not using PrEP expressed concerns regarding the potential side effects of PrEP, including long-term health impacts such as osteoporosis or kidney failure which deterred them from using PrEP as a self-protection option. A former PrEP user had stopped using PrEP within the last year partly due to concerns about potential side effects, "*I got scared because I started hearing that using PrEP could lead to kidney problems. So that's where I became scared and I didn't want to continue*" (participant #04, age 29, monogamous relationship). Another participant who had never used PrEP echoed similar sentiments stating, "*I have heard bad things before about drugs like PrEP. And I've never really known them to be true or untrue… And I've never had the opportunity to really speak to like a health care practitioner*" (Participant #07, age 32, polyamorous). Among the participants who had never used PrEP, side effect concerns coupled with the lack of access to reliable information decreased their willingness to use PrEP.

Participants primarily reported using PrEP due to their single status and active dating life or because they were in open relationships with multiple sexual partners. Additionally, one participant used PrEP due to being in a serodiscordant relationship (i.e., having a partner who is living with HIV). While PrEP users also expressed concerns regarding PrEP side effects, they felt that the HIV protection benefits outweighed the risk of long-term side effects. As shared by one participant, "*I'm like, oh the side effects of like, bone issues. Trust me, I'll rather be able to take care of my bones*" (Participant #12, age 40, open relationship). Similarly, the perspective that the potential benefits of PrEP outweigh the risks was shared by participant #12, a 34-year-old single male who had been diagnosed with kidney damage from PrEP use four years earlier. Despite this, he expressed a desire to explore safer PrEP options with his provider, citing the HIV protection benefits, "*I do want to get on PrEP and I do want to have sex. I'm not saying I'm going to have random hookups, but I feel it provides good security.*" Other participants highlighted that PrEP provided peace of mind, allowing them to enjoy better sexual experiences by reducing their fear and anxiety around HIV acquisition. As one participant expressed,

"creo que obviamente en el momento del encuentro es un poco más de pronto más placentero, pero es un poco más cómodo, en el sentido de que voy a sentir menos como presión o menos miedo [*I think that obviously, in the moment of the encounter, it's perhaps more pleasurable, but it's also a bit more comfortable in the sense that I feel less pressure or less fear…*]" (Participant #10, age 26, monogamous relationship)

These narratives highlighted that the perceived and experienced emotional and sexual health benefits helped to outweigh concerns about adverse side effects among PrEP users.

**Abstinence.** Notably, a minority of participants were practicing abstinence as an HIV self-protection strategy at the time of the interview. One participant described abstinence as his "*first line of defense*" (participant #03, age 24, single) against HIV. When discussing PrEP use, participants were cognizant that sometimes sex can be unplanned and sporadic but were unsure how to weigh the health risks of PrEP against those sporadic encounters. While not representative of the majority, these narratives highlight the range of HIV self-protection behaviors among Latinx SMM.

### Subtheme: HIV self-protection behaviors and decision-making dynamics

An important subtheme that emerged was that HIV self-protection varied with new partners compared to established partners. An established partner could be their significant other,

but for some participants, this also meant someone with whom they were casually dating or hooking up with regularly. Participants reported a heightened perception of HIV risk with new partners. As these participants stated,

> "Pero se que cada vez que se hace algo con otra persona osea, you know, you're out there. You're throwing an eight ball, you know? aver en donde cae, el que cae, pero eso que para mi importante este saber con quien estoy jugando con quien estoy haciendo las cosas [*But I know that every time something is done with another person, I mean, you know, you're out there. You're throwing an eight ball, you know? Seeing where it lands, who it lands on, but what's important for me is knowing who I'm playing with, who I'm doing things with*" (Participant #05, age 40, open relationship).

> "No te necesitas meterte con varias personas, no mas te metes con la persona equivocada. [*You don't need to get involved with multiple people, you just get involved with the wrong person*] (Participant #05, age 34, single).

The "*question mark*" around the HIV risk introduced by new partners led participants to practice caution when selecting new sexual partners.

Participants emphasized the importance of getting to know new partners and the role of open communication in mitigating HIV risk and informing their decisions about self-protection. They actively asked new partners about their sexual history, recent HIV testing, and condom use preferences, sexual positioning preferences (e.g., top, bottom, versatile) to gain a more accurate understanding of their sexual risk. These discussions also helped determine whether potential partners shared similar attitudes regarding sexual health. As one participant expressed,

> "Que yo veo que la otra persona no se interesa en usar condón no tiene no tiene interés de hacerse test eh es una mala señal una red flag y definitivamente eso no me daría la seguridad para estar con la otra persona. [*If I see is that if the other person is not interested in using a condom and has no interest in getting tested, it's a bad sign, a red flag, and definitely, that would not give me the confidence to be with that person.]*" (Participant #08, age 26, open relationship)

> "…that's why we'd rather do it [have sex] with people that we already know that are like us. That have been to the doctors, you know, that are the same as us, so we don't have to worry about that." (Participant #14, age 40, open relationship)

After screening partners, participants felt more informed to make decisions about whether to pursue a sexual relationship with the potential partner. Depending on the perceived level of risk, some participants opted for condom use with partners considered "*riskier,*" while others chose to engage in lower-risk sexual activities (e.g., mutual masturbation) or abstained from sex with those individuals altogether. However, participants also expressed skepticism about the accuracy of the information provided by new partners, as there was no way to verify its truthfulness. As one participant stated, "*I know what I have, and I don't have, and I know how I am and I know you know how healthy I am. But you know people can lie, and there's no proof or any hardcore evidence to say, hey, I don't have this*" (participant #13, age 29, single). This statement underscored the difficulties in balancing trust in a new partner and their sexual health choices. Participants also acknowledged that their decision-making process was not always straightforward; attraction to a person or certain situations (e.g., partying) could lead to unsafe encounters with unknown partners. Therefore, several participants emphasized the

importance of avoiding circumstances that could lead to sexual disinhibition, such as drug or alcohol use.

Open communication and sexual safety negotiating sexual safety with established partners centered around decisions made as a dyad to protect their sexual health. These topics included condom use preferences, frequency of HIV testing, PrEP use, sexual boundaries, and relationship agreements.

> "I think you know, when I first start conversations you know, the conversation always goes like have you taking your prep you know regularly." (Participant #06, age 26, single)

> "It's not necessarily putting a condom on. When people think you know safe sex; it's just understanding like when you have sex, who you're having it with, how you have it, and how you and the other person initiate and like understand like, what are the boundaries; how you're going to move forward; who's going to do what, etc." (Participant #12, age 34, single)

Importantly, individual acts of sexual health promotion and HIV self-protection took on the meaning of being an act of mutual protection between partners, whereby each partner shares the responsibility of protecting the other. Although mutual protection of sexual health between partners was more commonly brought up by participants in relationships, single participants also expressed that their acts of sexual health promotion and HIV self-protection are to protect others as well themselves. Most participants in monogamous relationships discussed having negotiated safety agreements with their partners who were living without HIV, where mutual protection included monogamy and routine HIV testing (e.g., every 6 months or annually) and did not use condoms. Moreover, participants in open relationships exemplified the way open communication and a commitment to mutual protection between partners can facilitate HIV self-protection. For instance, when describing their transition from monogamous to open relationship agreements with their partner, two participants described engaging extensive discussion with their partners about the boundaries and rules for engaging with other sex partners. These participants also made the proactive decision to use PrEP prior to introducing new partners into their relationships. It is important to note both participants were well educated and had access to healthcare, which they recognized as a privilege that other couples may not be afforded.

## Cultural and religious norms create a culture of silence around sexual health

The narratives of participants highlighted the profound influence of cultural and religious factors in shaping sexual health attitudes and behaviors. All participants grew up exposed to the social stigma associated with being a gay or bisexual man regardless of whether they grew up in the US or in Latin America. Specific cultural norms such as 'machismo', which emphasizes traditional male gender roles, challenged their ability to reconcile their sexual identities with societal expectations. As participant #14 (age 40, open relationship) stated, "*You know from the moment you proclaim yourself gay, that the male card gets taken away.*" This sentiment was echoed by others who highlighted the internal conflicts they experienced due to the clash between their sexual identities and the heteronormative standards imposed by the influence of religion in Latinx culture. One participant recounted,

> "So, my whole my whole upbringing was very uncomfortable because I wanted to pray or you know, to get the gay out, because I was raised in a home where that was wrong. You

know, men were to to get married and with a woman and have children and be you know, a fruitful, Christian young person." (Participant #02, age 40, open relationship)

Similarly, another participant recalled rejecting his feelings of same-sex attraction as early as 6 years old because "*[being gay] was not accepted*" (Participant #09, age 32, single) in his Christian family.

These sociocultural norms constrained the healthy development of sexual identity from childhood into young adulthood and hindered sexual health promotion behaviors. One participant shed light the profound impact of societal expectations in his life,

"….well my past experience right, I was you know as a straight man that was married for 20 years to a woman, three kids, Latino, Christian growing up right so you kind of lived the monogamous relationship, a Christian monogamous relationship and I did have my encounters outside, you know, because I was obviously, I've always been gay. I used to have those unfortunate encounters outside just to fulfill my sexual needs that I that I had, and they were not really safe… I was always afraid and scared that I would bring something to her right, and that would have been my you know, that was actually my worst nightmare." (Participant #14, age 40, open relationship)

This narrative of societal pressure and fear of stigma is echoed in the experience of another participant (age 32, polyamorous relationship) who shared being in committed relationships exclusively with women in his early twenties, despite knowing he was bisexual. He shared that machismo, religion, HIV stigma, and the added layer of biphobia (fear or hatred directed toward bisexual people) prevented him from coming out to his family and female partners.

"Coming from like the male side of the family and there is always a huge stigma about being gay… then my uncle who died of HIV. He's bisexual as well so there is an even larger stigma… So that that kind of like delayed me coming out of the closet big time." (Participant #07, age 32, polyamorous)

During this time, he was also having encounters with men outside of his relationships with women, often without protection or discussions about sexual health or HIV status.

While concealing their sexual identity protected against familial estrangement, it also led to feelings of anxiety and isolation. A poignant example is a 29-year-old, single participant recounted experiencing depression during his teenage years because he was concealing his sexual identity. After bravely coming out to his parents in high school, they reacted negatively and rejected him, which worsened his depression to the point of a suicide attempt.

"The coming out can be very traumatic, and you know, in my experience, like every other, I guess LGBT community gay person there's always depression and the thought of ending your life then and there…. mine was a very negative experience. I have attempted suicide. I have been through depression, mom and Dad took me to rehab for it. They were traumatized and then, you know, deep down they don't get that I'm rainbow but I am…"

He described the importance of coming out in his life not only in his relationship with his parents, but mostly in his relationship with himself,

"…when you come out and you have your support, your life just everything just blossoms and blooms. Clouds are parting with the rays of sunshine are coming down. The weight

off your shoulder has officially lifted. Your head doesn't weigh 1000 pounds. You can start focusing on other things like your career path or what you're gonna have for lunch tomorrow and whatnot… 'cause you do feel though the love and support afterwards, after your little like I said, you're at rock bottom." (Participant #13, age 29, single)

Many participants echoed similar initial reactions of rejection and disappointment from their families after coming out, but also regarded it as a difficult yet necessary experience towards greater self-acceptance. Embracing and openly owning their sexual identity provided them with the space to start prioritizing their sexual health.

Although most participants reported improved acceptance of their sexual identity by their families over time and strong family bonds, their narratives highlighted the ongoing role of families in perpetuating stigma post-coming out. For instance, some participants continued to conceal certain aspects of their sexual identity, like their dating life, to avoid negative or judgmental reactions. For example, participant #15 (age 40, single), out to his family for nearly 20 years, shared that his parents and sisters often make veiled homophobic comments, such as asking if he's going to meet "*amigos de los tuyos [friends like you]*" when they suspect he's seeing another man. Similarly, another participant recounted how his mother heightened the stigma around HIV when he came out as gay in his teens, warning him, "*If you are sleeping around with these men, you're going to get HIV* " (participant #03, age 24, single). Despite the stigmatizing nature of her comments, he recognized they came from a place of protection and credited his mother's warnings for his diligent HIV self-protection practices. Interestingly, a few other participants described that this stigma fueled their motivation to protect themselves from HIV. Their stories illustrated how family-induced stigma influenced their self-acceptance and shaped their HIV self-protective behavior, before and after coming out.

## Lack of information and misinformation leading to self-reliance for sexual health protection

Across the interviews, a recurring theme in participant narratives was the absence of comprehensive sexual health education during their formative years. Many participants did not receive any sexual health education from their parents and did not feel comfortable discussing these topics with them because of the silence around sex, including sexual identity, prevalent in Latinx communities. Importantly, participants had received limited or no formal sexual health education in school regardless of whether they went to school in the U.S. or in Latin America, which highlighted that sexual conservatism is not only a cultural manifestation but a broader societal issue. Participants felt that sexual health conversations with their parents could have potential to help them be more informed. The absence of comprehensive sexual health education led to participants engaging in sexual exploration without a full grasp of how to protect their sexual health and against HIV. This lack of knowledge often resulted in periods of increased risk, particularly during adolescence and young adulthood, as participants navigated their sexual experiences without the necessary knowledge to effectively engage in HIV self-protection. As participant #03 (age 24, single) reflected, "*I think, looking back, there were times when I was definitely at higher risk because I just didn't know any better. It's like you're trying to figure it all out on your own, and that can be really dangerous.*" This sentiment was shared by other participants who in absence of formal sexual education, learned lessons about sexual health promotion through trial and error, such as acquiring their first STI. Unfortunately, for some participants, STI treatment was also their first opportunity to engage in a sexual health discussion with a provider.

The lack of sexual health education made it difficult for Latinx SMM to parse through the misinformation that exists around sexual health. As one participant described,

> "Oh, pues siento que, al principio bueno, desde el primer momento en que empecé a hacer activo sexualmente, pienso que la protección que uno más tiene es el miedo.¿Por qué? La desinformación; entonces al principio, pues no es mucha la información que uno recibe de pronto más en el país del que yo vengo, como que en las escuelas no se habla de mucha educación sexual; de pronto se viene más de la familia, pero a veces es como muy limitada la información. Entonces, pues al principio es como el miedo en cuanto a muchas cosas,¿no? Entonces pienso que es una manera de cuidarte es bueno sintiendo miedo. [*Oh, well I feel that, at first, well, from the first moment I started being sexually active, I think the greatest protection one has is fear. Why? Because of misinformation; so at the beginning, there isn't much information, especially in the country I come from, where there isn't much talk of sexual education in schools; it often comes more from the family, but sometimes the information is very limited. So, at the beginning, it's like fear about many things, right? So, I think that one way to take care of yourself is by feeling fear.*]" – (Participant #10, age 26, monogamous relationship)

Despite this lack of sexual health education, many participants had taken a proactive approach to become educated about sexual health and HIV in adulthood by actively seeking out reliable sources of information. Even though this process was often described as intimidating, they sought information from various sources including from healthcare providers, online resources, or community organizations. For example, one participant shared his personal experience seeking sexual health care for the first time,

> "My father never talked to me about, you know, sexually transmitted diseases or anything like that. My mom was a little open, in the sense that, you know, you got to practice abstinence, but, you know, never did they give me a condom, or did they give me anything like that. And since a very young age that I was in [city name] at 21, I did go to like, the human services to make sure I got tested for HIV, or any, you know, anything like that. And it's a little intimidating when you don't know exactly what you want, you know, what you're going into, but at the same time, you know, it broke me. And they're like, here's your condoms, here's what you need, and move forward." (Participant #02, age 40, open relationship)

The proactive efforts in seeking reliable sexual health information taken by participants reflected their resilience in overcoming educational deficiencies and in taking ownership of their sexual health.

## Growing wiser: maturity's impact on sexual health and relationships

Participants' narratives revealed a core theme of maturity, reflecting their journeys toward more self-protective behaviors, and overcoming challenges related to limited awareness and stigma. As individuals navigated the complexities of accepting their sexual identity and establishing themselves professionally within their communities, participants described a stronger sense of self-worth with these milestones, which gave them the confidence to take ownership of their sexual health.

> "Just like being out to like my parents and communities and just overall, like coworkers and I feel like there's a sense of maturity from when I first came out to like where I am

now, and so I think quote, unquote, being an adult like now, I think I'm able to, you know, take that accountability and take that confidence in saying, you know, like if you if you can't respect to my values and my control of my sexual health and you know, like maybe, I shouldn't be you know interacting with you or like speaking on this action that shouldn't be taken lightly." (Participant #06, age 27, single)

This included setting healthy boundaries with partners, choosing partners whose values align with their own, and thinking beyond physical desires when making decisions about partners.

Latinx SMM also discussed how sexual experiences and previous relationships influenced them to take better care for their sexual health. A participant described that sexual experiences made him wiser, allowing him to detect "*red flags*" from partners, which he perceived as giving him more control over his sexual health.

"As you become more, I guess mature in age you start... things start clicking in your mind. You know what's a pinpoint suspicious attitude or I guess energies and personalities like ooh no just that right there was... that's a big red flag and red is not a super color, and you only get more wiser with experience or with stories." (Participant #13, age 29 single)

With wisdom and maturity, participants reported becoming more selective in their engagement with potential partners, favoring "quality" over "quantity" in their relationships. This selectiveness was driven by a better understanding of their own needs and their ability to communicate those needs with partners. Several participants described a notable shift towards valuing deeper emotional connections over purely physical or superficial attractions.

"… like other things are important to me. You know, looks very important in my 20s and there still are, you know you definitely want a good-looking partner; But to me I think what's most important now is how someone makes me feel, what they bring to the table. No matter what they look like but am I at risk...? …When I lived in New York, the first like of the first three or four years, I would basically hook up like multiple times on a week…" (Participant #12, age 34, single)

These narratives reflected not just a change in sexual behaviors, but a deeper understanding of what truly fulfills them in interpersonal connections.

### Humanizing unjust sexual health care

Participants provided descriptions of unjust and dehumanizing healthcare experiences, but they also provided glimpses of hope for a more human experience. Healthcare provider attitudes and the healthcare environment had an impact on how participants accessed sexual healthcare. A common thread across participant experiences was feeling judged or stigmatized by healthcare providers. This stigma was often related to their sexual orientation, sexual behavior, or their health concerns, such as STI testing or treatment. A participant felt he had been reprimanded and 'scolded' by the nurses when seeking care at his local health department after being exposed to syphilis. Latinx SMM also anticipated and experienced stigma from healthcare providers related to PrEP use. Some healthcare providers have limited familiarity with PrEP and even suggested discouraged the use of PrEP. One participant, shared his experience,

"I've had difficulty with doctors and putting their own opinion or even initiating the conversation. Just the stigma of like of the sexual behaviors and the doctor is not feeling

comfortable with like prescribing the drug because they feel like it could be preventable if you just don't engage in those activities." (Participant #06, age 27, single).

These negative experiences with healthcare providers led some Latinx SMM to feel reluctant to access sexual healthcare and discouraged open communication about sexual health.

In contrast, providers who were LGBTQ+ affirming, non-judgmental, and knowledgeable facilitated sexual health promotion for Latinx SMM and were actively sought out by participants.

> "I would always look for. Like a younger female Doctor who is like more like him to meet with the gay world, right? Or like. Or a doctor that says they're gay, friendly and like advertises on like gay websites you know, or gay directories and stuff like that. Just because you know we want to be 100% completely open with your health care practice, right?" (Participant #07, age 32, polyamorous)

Several participants articulated positive experiences of engaging in shared decision-making with their healthcare providers because they approached sexual healthcare in a respectful and understanding manner. Providers were also helpful in helping participants address any concerns about beginning PrEP, by providing information about side effects and reassurance about benefits of PrEP use. Additionally, healthcare providers who delivered kind and compassionate care helped to mitigate negative aspects of the healthcare environment for Latinx SMM.

> "Sometimes I feel that if you don't go to like these public places for service, they dehumanize you, you know? They make you like, ugh you can't afford it and look at you, you're, you're insecure. I don't know, I felt dehumanized by others. But she humanized me. She was very nice, no matter when I had a call whatever. She was always, she was always like, how's the prep, um, let's go get your blood work, you know, she was always on top of things, making sure that I was okay." (Participant #02, age 40, open relationship)

Regardless of whether participants had an established relationship with a healthcare provider, they consistently cited providers as trusted sources of sexual health information and recognized the role they play in sexual health promotion. Normalizing sexual health care through advertising on social media and in the community as well as increasing visibility of sexual health clinics was described as important for stigma reduction and increasing access to care among Latinx SMM.

Financial barriers and lack of health insurance were also significant structural barriers to accessing care for Latinx SMM. Participants in this study revealed that financial barriers had impeded their access to sexual health care at some point in their lives and cited high costs of healthcare visits as a deterrent to seeking healthcare. Additionally, participants identified the lack of health insurance as a critical barrier to accessing sexual health care, particularly among self-employed individuals. The high costs of PrEP were cited by participants as a significant obstacle to its use, irrespective of their health insurance status. These narratives highlighted several structural barriers that are known to limit Latinx SMM's access to sexual healthcare and HIV self-protection options.

## Discussion

This qualitative study contributes uniquely to the literature by expanding the understanding of the sociocultural barriers and facilitators of sexual health promotion among Latinx SMM.

Key findings in this study highlight HIV prevention as a priority within this community despite profound challenges they face accessing sexual health care. This study also uncovered the dynamic nature of how Latinx SMM employ HIV self-protection strategies which were shaped by individual preferences and partner dynamics. Additionally, this study underscores the impact of intersecting identities, such as being Latinx and SMM, on experiences of stigma, access to services, and sexual health outcomes. Primarily, how the lack of access to sexual health education undermines the sexual health of Latinx SMM. This study also shed light on how Latinx SMM perceive that their maturity and life experiences enabled them to care for their sexual health, including the prevention of HIV.

An important finding of this study was that HIV self-protection behaviors were dynamic and varied across participants. This finding echoes findings from previous research that the HIV self-protection strategies employed by SMM vary within and across sexual partners or experiences [37–39]. This research highlights how SMM have adapted their sexual behaviors during the HIV epidemic, incorporating strategies such as serosorting and biomedical matching in response to advancements in treatment and prevention [39]. Serosorting involves individuals choosing sexual partners based on their HIV status, while biomedical matching refers to decisions (e.g., condom use) made based on the use of PrEP or viral suppression in HIV-positive partners [37,39]. Findings from this study reflect these patterns and highlighted the influence of multiple factors, including individual beliefs, relationship dynamics, and access to resources, on decision-making processes related to HIV self-protection. There is an urgent need to support informed HIV self-protection decision-making among Latinx SMM through the development of more flexible and adaptable HIV prevention interventions to address their sexual health needs.

This study illustrates the profound impact the pervasive stigmas surrounding HIV, on the sexual health behaviors of Latinx SMM. Notably, the homophobia, HIV-stigma, and sexual silence within their families led to internalized homophobia, which has been shown to hinder sexual health promotion behaviors [40,41]. Machismo reinforces these stigmas and exacerbates barriers to sexual health promotion, including HIV self-protection behaviors such as PrEP use [42,43]. This sociocultural stigma also had a negative impact on the mental health of Latinx SMM, who feared being ostracized by their families. Consistent with other research, Latinx SMM reported that their family's acceptance of their sexual identity improved with time [44]. However, because of the multiple stigmas attached to their sexual identity they choose to keep certain aspects of their life private, such as their dating life. While this was described as protective for themselves and their families, there is a dearth of research exploring how quality of familial relationships impact the health of Latinx SMM. Given that the importance of family remains central for Latinx SMM despite challenges around sexual identity [45], future research should explore how openness with family, beyond matters of sexual identity, influences health outcomes in this group.

The lack of sexual education and accessible information on sexual healthcare resources reported by Latinx SMM in this study highlight ongoing challenges identified in prior research [46–48]. For example, studies on PrEP among Latinx SMM highlight gaps in PrEP knowledge and resources that significantly hinder their ability to engage in PrEP use [49,50]. The absence of sexual health education in schools and from parents left participants without the guidance they needed to protect their sexual health as they became sexually active. A growing body of literature has underscored the important role parents play in sexual health education among sexual and gender minoritized youth, including Latinx SMM [51–55]. Our findings that Latinx SMM would have appreciated sexual health discussions with their parents further support the role of parental involvement in bolstering sexual health outcomes among this group. Given that Latinx SMM aged 13–35 have the highest rates of new STI and HIV

infections in the U.S. [2], addressing gaps in sexual health education through both formal education settings and family-based discussions is critical for improving sexual health outcomes among Latinx SMM..

Our findings also highlighted the relationship between maturity and sexual health among Latinx SMM. Maturity, in this context, involved not only aging but also learning skills for protecting their sexual health from sexual experiences. For instance, many participants developed self-advocacy skills, enabling them to navigate structural barriers, seek out resources, and access necessary sexual healthcare. In addition, they mentioned the broader development of self-awareness, communication skills, and understanding of sexual health risks and HIV self-protection practices. While learning from sexual experiences is an inherent aspect of sexual development, Latinx SMM need resources to navigate these experiences safely. Peer sexual health education interventions have been demonstrated to have a positive impact sexual health promotion and HIV self-protection outcomes [56]. These interventions can ensure that essential knowledge and practices for safe sexual behavior are communicated in safe and affirming environments. Findings from this study highlight the need for strategies to support and empower Latinx SMM with the necessary resources, such as peer interventions, to safely navigate their sexual health journey.

Our participants highlighted structural barriers in accessing sexual healthcare, including financial constraints and insurance limitations. These barriers often impede the ability of Latinx SMM to seek timely and appropriate sexual health services [46,57]. Additionally, participants reported mixed experiences with healthcare providers, with some indicating that providers were not always supportive of their sexual healthcare. This inconsistency in care can lead to mistrust and reluctance to seek healthcare services, further exacerbating sexual health inequities [58]. These findings underscore the need for targeted interventions to address financial and insurance barriers, as well as the importance of training healthcare providers in culturally responsive care practices.

## Limitations

These study findings should be interpreted within the context of the study's limitations. The study's findings are derived from a limited number of interviews of Latinx SMM living in a handful of states, which may not fully represent the experiences of those in other settings. All participants reported ancestry from Latin American countries and were either Spanish- or English-speaking. As a result, our sampling approach inherently excluded non-Spanish-speaking members of the Latinx community, including individuals who primarily speak indigenous languages, Portuguese, or other languages. This limitation restricts the generalizability of our findings to the broader Latinx population, only captured the experiences of Spanish- and English-speaking Latinx individuals. Furthermore, some of the data cited in this study use the term 'Hispanic,' which typically refers to Spanish-speaking populations and primarily reflects data from Spanish- and English-speaking individuals. Therefore, it is difficult to ascertain whether the findings of this study align more broadly with those reported in the cited literature. Future research should aim to include greater linguistic and cultural diversity to more accurately reflect the full spectrum of the Latinx community and provide a more comprehensive understanding of their experiences.

Additionally, data collection began in October of 2020 during the COVID-19 pandemic at a time when there was significant uncertainty and rapidly changing information regarding transmission, prevention, and treatment of COVID-19. This evolving context could have influenced participants' perceptions and behaviors related to sexual health and HIV self-protective actions, potentially making them more cautious or altering their usual practices

due to fears of COVID-19 transmission. Stringent social and physical distancing measures implemented early in the pandemic likely affected participants' social interactions and opportunities for sexual encounters. These measures could have temporarily influenced sexual behaviors, potentially reduced the incidence of new sexual partnerships, and thereby impacted the relevance of certain HIV self-protective behaviors.

## Conclusion

In conclusion, this qualitative study sheds light on the intricate interplay of sociocultural factors and their impact on sexual health promotion and HIV self-protection behaviors among Latinx SMM. These findings emphasize the need for focus on addressing structural barriers such as sexual health education, access to care, and healthcare provider bias to improve sexual health outcomes among Latinx SMM. Given the wide range of HIV self-protection options available today, there is a need to develop interventions that will support informed HIV self-protection decision-making among Latinx SMM. Additionally, future research should explore the role of family and peer networks in enhancing the effectiveness of these interventions. By leveraging the strengths within the Latinx SMM community, we can create more impactful strategies for promoting sexual health and HIV self-protection.

## Author contributions

**Conceptualization:** Lisvel A. Matos, Jasmine Salas, Caroline Munoz, Rosa Gonzalez-Guarda.

**Data curation:** Lisvel A. Matos, Jasmine Salas, Caroline Munoz, Rosa Gonzalez-Guarda.

**Formal analysis:** Lisvel A. Matos, Sarah E. Janek, Jasmine Salas, Caroline Munoz, Michael V. Relf, Rosa Gonzalez-Guarda.

**Funding acquisition:** Lisvel A. Matos.

**Investigation:** Lisvel A. Matos.

**Methodology:** Lisvel A. Matos, Rosa Gonzalez-Guarda.

**Project administration:** Lisvel A. Matos.

**Resources:** Lisvel A. Matos.

**Software:** Lisvel A. Matos.

**Supervision:** Lisvel A. Matos.

**Validation:** Lisvel A. Matos, Sarah E. Janek, Michael V. Relf, Rosa Gonzalez-Guarda.

**Visualization:** Lisvel A. Matos.

**Writing – original draft:** Lisvel A. Matos, Sarah E. Janek, Michael V. Relf, Rosa Gonzalez-Guarda.

**Writing – review & editing:** Lisvel A. Matos, Sarah E. Janek, Jasmine Salas, Caroline Munoz, Michael V. Relf, Rosa Gonzalez-Guarda.

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
