## [Decision Letter · Decision Letter 0]

19 Nov 2024

PONE-D-24-42433Understanding Sociocultural Influences in Sexual Health Promotion and HIV Protection among Latinx Sexually Minoritized Men: A Qualitative StudyPLOS ONE

Dear Dr. Matos,

Thank you for submitting your manuscript to PLOS ONE. After careful consideration, we feel that it has merit but does not fully meet PLOS ONE’s publication criteria as it currently stands. Therefore, we invite you to submit a revised version of the manuscript that addresses the points raised during the review process. In regards to reviewer observations about clarity in terms of defining the population, please consider using the suggested definition of "only Spanish-speaking sub-population of Latinx" with an acknowledgement of the diversity of cultures and languages spoken in the Latine community.

We look forward to receiving your revised manuscript.

Kind regards,

Jonathan Garcia

Academic Editor

PLOS ONE

Journal Requirements:

2. We noted in your submission details that a portion of your manuscript may have been presented or published elsewhere. This manuscript was included in my dissertation and submitted to PROQuest. The dissertation has been embargoed until July 2026 and is therefore not publically available.

 Please clarify whether this conference proceeding or publication was peer-reviewed and formally published. If this work was previously peer-reviewed and published, in the cover letter please provide the reason that this work does not constitute dual publication and should be included in the current manuscript.

Reviewers' comments:

Reviewer's Responses to Questions

**Comments to the Author**

1. Is the manuscript technically sound, and do the data support the conclusions?

Reviewer #1: Yes

Reviewer #2: Yes

2. Has the statistical analysis been performed appropriately and rigorously? 

Reviewer #1: Yes

Reviewer #2: N/A

3. Have the authors made all data underlying the findings in their manuscript fully available?

Reviewer #1: No

Reviewer #2: Yes

4. Is the manuscript presented in an intelligible fashion and written in standard English?

Reviewer #1: Yes

Reviewer #2: Yes

5. Review Comments to the Author

Reviewer #1: Thank you for the opportunity to review this manuscript. The paper is well-written, and the authors provide incredibly rich data and a very thoughtful analysis. The findings offer important insights into what shapes sexual health behavior among Latinx SMM, which is highly useful for informing future research and intervention efforts. With some minor revisions (comments below), it will make a valuable contribution to PLOS ONE and the public health field.

Abstract:

- Lines 17-18: Suggest adding 1-2 examples of self-protective behaviors in parentheses

- Lines 19-20: Recruitment strategy should be (briefly) indicated here (e.g., an online convenience sample of...)

- Lines 21-22: It looks like this sentence is meant to justify the age of the data, which doesn't seem necessary given it's relatively recent; it is also redundant with the material above, so it could be removed (unless required by the journal)

- Suggest adding a sentence about implications of your findings (e.g., how it might inform future research or intervention)

Introduction:

- Lines 52-65: The authors nicely summarize the literature on barriers to sexual health access among Latinx SMM, including sociocultural factors. However, it is not entirely clear how this study addresses pertinent gaps (the authors state this in lines 65-67, more but specificity is needed). Please revise to say more about the types of gap(s) that this study addresses (e.g., methodological gap; focus on certain HIV self-protective behaviors or barriers/facilitators that have received less attention...)

Methods:

- Suggest moving the Data Collection section up, after Setting and before Study Materials

- Data Collection section: It is stated that the interview was semi-structured, but it would be helpful to have more information on the types of interview questions/domains (e.g., broadly asked about sexual health promotion vs. specific HIV/STI prevention behaviors). This could include a few sample questions (structured or open-ended) and/or follow-up probes.

- Lines 100-101: This sentence is a little unclear in describing the number of participants; suggest rephrasing to "Of those, 15 individuals enrolled in the study, including..."

- Line 144: It is unclear what is meant by "...in batches balancing recruitment, data collection, and analyses". Please rephrase.

- Lines 144-146: The authors note that new codes emerged in analyzing the latter 10 interviews; please state whether these codes were applied/back-coded to the first 5 cases that were coded

Results:

- Pg. 13, 1st paragraph: Did discussions of HIV include the type/source of testing (e.g., testing in a clinic or HIV self-testing). If so, please indicate.

- Pgs. 12-15: This theme is very rich and there is a lot of information in this section; suggest adding sub-headers to help organize (e.g., HIV testing, condom use, PrEP, other)

- The authors describe similarities regarding minimal sex education received by participants who grew up in the US vs. Latin America. Did any other similarities/differences emerge between participants who immigrated to the US vs. born in the US?

Discussion:

- Throughout the manuscript but particularly in the discussion, there are some run-on sentences and minor grammar errors. Suggest further proofreading and using more concise language when possible.

Reviewer #2: Thank you for the opportunity to review this article titled, "Understanding Sociocultural Influences in Sexual Health Promotion and HIV Protection among Latinx Sexually Minoritized Men: A Qualitative Study". The authors conducted qualitative interviews with the intention of gaining insights into the barriers to PrEp uptake among sexual minoritized Latinx men. This article is well written and the data contributes meaningfully to the literature. However, I do have major concerns regarding how the study population is defined which impacts how the data is interpreted and how it can be generalized.

Authors appear to use the terms “Hispanic” and “Latinx” interchangeably, they are NOT interchangeable terms. This centers Spanish as “the” language of the Latine which is not only inaccurate, it is harmful. There is already a term available to refer to Spanish-speaking peoples, co-opting “Latinx” for this same purpose is an act of oppression against those Latine whose ancestry is not linked to the Spanish language.

Some of the statistics cited are in reference to “Hispanic” people, yet the authors nevertheless report that they refer to “Latinx” communities in there narrative - again, these two populations are not defined in the same way, these terms cannot be used interchangeably. It is an act of erasure of non-Spanish speaking Latine communities to superimpose the term “Latinx” on data that refers only to Spanish-speaking peoples.

The authors define their study population as Latinx however they sampled only individuals who speak Spanish or English, from a longitudinal study of young adult Hispanic immigrants in the U.S, and on the internet. All study participants report ancestry from Spanish-speaking countries in the Americas/Caribbean. By definition, the Latinx population also includes individuals from countries in the Americas/Caribbean where Portuguese and French are spoken. It would be more accurate to refer to the study population as “Spanish-speaking” or “Hispanic” or to otherwise specify “only Spanish-speaking sub-population of Latinx", as these findings cannot reasonably be generalized to portions of the Latine population in the U.S. that were systematically excluded from participation in this study.

The authors did not report the exclusion criteria.

6. PLOS authors have the option to publish the peer review history of their article (what does this mean? ). If published, this will include your full peer review and any attached files.

**Do you want your identity to be public for this peer review?** For information about this choice, including consent withdrawal, please see our Privacy Policy .

Reviewer #1: **Yes: ** Ashley Schuyler

Reviewer #2: No

---

## [Author Response · Author response to Decision Letter 1]

3 Jan 2025

We noted in your submission details that a portion of your manuscript may have been presented or published elsewhere. This manuscript was included in my dissertation and submitted to PROQuest. The dissertation has been embargoed until July 2026 and is therefore not publically available.

Please clarify whether this conference proceeding or publication was peer-reviewed and formally published. If this work was previously peer-reviewed and published, in the cover letter please provide the reason that this work does not constitute dual publication and should be included in the current manuscript.

Explanation provided in cover letter.

We note that you have indicated that there are restrictions to data sharing for this study. For studies involving human research participant data or other sensitive data, we encourage authors to share de-identified or anonymized data. However, when data cannot be publicly shared for ethical reasons, we allow authors to make their data sets available upon request. For information on unacceptable data access restrictions, please see http://journals.plos.org/plosone/s/data-availability#loc-unacceptable-data-access-restrictions

If there are ethical or legal restrictions on sharing a de-identified data set, please explain them in detail (e.g., data contain potentially identifying or sensitive patient information, data are owned by a third-party organization, etc.) and who has imposed them (e.g., a Research Ethics Committee or Institutional Review Board, etc.). Please also provide contact information for a data access committee, ethics committee, or other institutional body to which data requests may be sent.

If there are no restrictions, please upload the minimal anonymized data set necessary to replicate your study findings to a stable, public repository and provide us with the relevant URLs, DOIs, or accession numbers. Please see http://www.bmj.com/content/340/bmj.c181.long for guidelines on how to de-identify and prepare clinical data for publication. For a list of recommended repositories, please see https://journals.plos.org/plosone/s/recommended-repositories. You also have the option of uploading the data as Supporting Information files, but we would recommend depositing data directly to a data repository if possible.

A minimal anonymized data set has been uploaded to QDR.org. This explanation has been provided in the cover letter and the Data Availability Statement has been updated to reflect this change.

Reviewer #1: Thank you for the opportunity to review this manuscript. The paper is well-written, and the authors provide incredibly rich data and a very thoughtful analysis. The findings offer important insights into what shapes sexual health behavior among Latinx SMM, which is highly useful for informing future research and intervention efforts. With some minor revisions (comments below), it will make a valuable contribution to PLOS ONE and the public health field.

Abstract:

- Lines 17-18: Suggest adding 1-2 examples of self-protective behaviors in parentheses

Thank you for this suggestion. We have addressed this by adding examples of self-protective behaviors in the abstract on lines 18-19.

- Lines 19-20: Recruitment strategy should be (briefly) indicated here (e.g., an online convenience sample of...)

We have addressed this comment by including a brief description of the recruitment strategy in the abstract on lines 21-22.

- Lines 21-22: It looks like this sentence is meant to justify the age of the data, which doesn't seem necessary given it's relatively recent; it is also redundant with the material above, so it could be removed (unless required by the journal)

Thank you for your feedback. We agree that this sentence is not necessary and have removed it from the abstract as suggested.

- Suggest adding a sentence about implications of your findings (e.g., how it might inform future research or intervention)

We have addressed this by adding a sentence about the implications of our findings in the abstract on lines 27-29.

Introduction:

- Lines 52-65: The authors nicely summarize the literature on barriers to sexual health access among Latinx SMM, including sociocultural factors. However, it is not entirely clear how this study addresses pertinent gaps (the authors state this in lines 65-67, more but specificity is needed). Please revise to say more about the types of gap(s) that this study addresses (e.g., methodological gap; focus on certain HIV self-protective behaviors or barriers/facilitators that have received less attention...)

We appreciate this suggestion. We have revised the manuscript on lines 70-74 to provide greater specificity about the types of gaps this study addresses, including the focus on less-explored HIV self-protective behaviors and barriers/facilitators unique to Latinx SMM.

Methods:

- Suggest moving the Data Collection section up, after Setting and before Study Materials

We have made this change and moved the Data Collection section to follow the Setting section on lines 124-137.

- Data Collection section: It is stated that the interview was semi-structured, but it would be helpful to have more information on the types of interview questions/domains (e.g., broadly asked about sexual health promotion vs. specific HIV/STI prevention behaviors). This could include a few sample questions (structured or open-ended) and/or follow-up probes.

We appreciate this suggestion. We have expanded the Data Collection section to include examples of the interview questions and follow-up probes used to guide discussions on lines 129-132, as requested.

- Lines 100-101: This sentence is a little unclear in describing the number of participants; suggest rephrasing to "Of those, 15 individuals enrolled in the study, including..."

We have corrected this sentence for clarity as recommended on lines 114-119.

- Line 144: It is unclear what is meant by "...in batches balancing recruitment, data collection, and analyses". Please rephrase.

We agree that it was confusing and have rephrased this sentence for clarity on lines 164-165.

- Lines 144-146: The authors note that new codes emerged in analyzing the latter 10 interviews; please state whether these codes were applied/back-coded to the first 5 cases that were coded Thank you for this important feedback. Yes, the new codes that emerged during the analysis of the latter 10 interviews were applied and back-coded to the initial five cases.

We have clarified this point in the manuscript on lines 166-168.

Results:

- Pg. 13, 1st paragraph: Did discussions of HIV include the type/source of testing (e.g., testing in a clinic or HIV self-testing). If so, please indicate.

Thank you for this question. We did not specifically ask participants about HIV self-testing during the interviews. However, in their responses, participants often referred to going to a clinic or seeing a healthcare provider for HIV testing. We have clarified this point in the manuscript on lines 248-249.

- Pgs. 12-15: This theme is very rich and there is a lot of information in this section; suggest adding sub-headers to help organize (e.g., HIV testing, condom use, PrEP, other)

Thank you for your suggestion. We have added sub-headers (e.g., HIV testing, condom use, PrEP, other) to this section to help organize the content, as recommended, on pages 13-16.

- The authors describe similarities regarding minimal sex education received by participants who grew up in the US vs. Latin America. Did any other similarities/differences emerge between participants who immigrated to the US vs. born in the US?

The other theme where similarities between participants who immigrated to the US and those born in the US emerged was in the cultural and religious norms theme. With both groups highlighting that cultural expectations and religious beliefs significantly influenced their sexual health behaviors and attitudes. We have added a sentence to the manuscript on lines 390-392 to highlight this similarity.

Discussion:

- Throughout the manuscript but particularly in the discussion, there are some run-on sentences and minor grammar errors. Suggest further proofreading and using more concise language when possible.

Thank you for noting these issues. We have thoroughly proofread the manuscript and revised the discussion section, as well as the rest of the text, to eliminate run-on sentences and improve conciseness.

Reviewer #2:

Reviewer #2: Thank you for the opportunity to review this article titled, "Understanding Sociocultural Influences in Sexual Health Promotion and HIV Protection among Latinx Sexually Minoritized Men: A Qualitative Study". The authors conducted qualitative interviews with the intention of gaining insights into the barriers to PrEp uptake among sexual minoritized Latinx men. This article is well written and the data contributes meaningfully to the literature. However, I do have major concerns regarding how the study population is defined which impacts how the data is interpreted and how it can be generalized.

Authors appear to use the terms “Hispanic” and “Latinx” interchangeably, they are NOT interchangeable terms. This centers Spanish as “the” language of the Latine which is not only inaccurate, it is harmful. There is already a term available to refer to Spanish-speaking peoples, co-opting “Latinx” for this same purpose is an act of oppression against those Latine whose ancestry is not linked to the Spanish language.

Thank you for bringing this important issue to our attention. We have revised the manuscript to use the term "Latinx" consistently throughout when referring to the populations discussed. Additionally, we have clarified that our study focuses on a Spanish-speaking sub-population of Latinx individuals and explicitly acknowledge that this does not encompass the full linguistic and cultural diversity of the Latinx community. These changes can be found on lines 32-35, 87-88, and 97-99.

Some of the statistics cited are in reference to “Hispanic” people, yet the authors nevertheless report that they refer to “Latinx” communities in there narrative - again, these two populations are not defined in the same way, these terms cannot be used interchangeably. It is an act of erasure of non-Spanish speaking Latine communities to superimpose the term “Latinx” on data that refers only to Spanish-speaking peoples.

Thank you for raising this important point. We acknowledge the distinction between the terms "Hispanic" and "Latinx" and recognize that they are not interchangeable. In our manuscript, we have chosen to use the term "Latinx" consistently to refer to the populations discussed, as it is a gender-neutral term intended to encompass individuals of Latin American ancestry. We provide this definition on lines 32-35.

While some of the statistics cited in our work use the term "Hispanic," we use "Latinx" in our narrative to align with our study’s focus on inclusivity and to reflect the terminology commonly used in current research and discourse.

We revised the manuscript to specify that our use of Latinx throughout the manuscript refers to Spanish- and English speakers on lines 87-89. We have also expanded the limitations of our study’s inclusion criteria and cited research on lines 668-678.

The authors define their study population as Latinx however they sampled only individuals who speak Spanish or English, from a longitudinal study of young adult Hispanic immigrants in the U.S, and on the internet. All study participants report ancestry from Spanish-speaking countries in the Americas/Caribbean. By definition, the Latinx population also includes individuals from countries in the Americas/Caribbean where Portuguese and French are spoken. It would be more accurate to refer to the study population as “Spanish-speaking” or “Hispanic” or to otherwise specify “only Spanish-speaking sub-population of Latinx", as these findings cannot reasonably be generalized to portions of the Latine population in the U.S. that were systematically excluded from participation in this study.

We acknowledge that our study sampled only Spanish- and English-speaking individuals with ancestry from Spanish-speaking countries in the Americas and Caribbean. To address this, we have revised the manuscript to more accurately describe our study population as a Spanish- and English speaking sub-population of Latinx. Additionally, we have clarified in the limitations section that our sampling approach excluded individuals from non-Spanish-speaking communities within the Latinx population on lines 673-676.

The authors did not report the exclusion criteria.

Thank you for pointing this out. We have added the exclusion on lines 102-103.

---

## [Editor Report · Decision Letter 1]

10 Jan 2025

Understanding Sociocultural Influences in Sexual Health Promotion and HIV Protection among Latinx Sexually Minoritized Men: A Qualitative Study

PONE-D-24-42433R1

Dear Dr. Matos,

We’re pleased to inform you that your manuscript has been judged scientifically suitable for publication and will be formally accepted for publication once it meets all outstanding technical requirements.

Kind regards,

Jonathan Garcia

Academic Editor

PLOS ONE
---

## [Editor Report · Acceptance letter]

PONE-D-24-42433R1

PLOS ONE

Dear Dr. Matos,

I'm pleased to inform you that your manuscript has been deemed suitable for publication in PLOS ONE. Congratulations! Your manuscript is now being handed over to our production team.

Kind regards,

on behalf of

Dr. Jonathan Garcia

Academic Editor

PLOS ONE